

# Uncovering the microbiome of invasive sympatric European brown hares and European rabbits in Australia

Somasundhari Shanmuganandam[1,2], Yiheng Hu[1], Tanja Strive[2,3], Benjamin Schwessinger[1] and Robyn N. Hall[2,3]

[1] Research School of Biology, Australian National University, Acton, ACT, Australia
[2] Health & Biosecurity, Commonwealth Scientific and Industrial Research Organisation, Acton, ACT, Australia
[3] Centre for Invasive Species Solutions, Bruce, ACT, Australia

## ABSTRACT

**Background:** European brown hares (*Lepus europaeus*) and European rabbits (*Oryctolagus cuniculus*) are invasive pest species in Australia, with rabbits having a substantially larger environmental impact than hares. As their spatial distribution in Australia partially overlaps, we conducted a comparative microbiome study to determine how the composition of gastrointestinal microbiota varies between these species, since this may indicate species differences in diet, physiology, and other internal and external factors.

**Methods:** We analysed the faecal microbiome of nine wild hares and twelve wild rabbits from a sympatric periurban reserve in Canberra, Australia, using a 16S rRNA amplicon-based sequencing approach. Additionally, we compared the concordance between results from Illumina and Nanopore sequencing platforms.

**Results:** We identified significantly more variation in faecal microbiome composition between individual rabbits compared to hares, despite both species occupying a similar habitat. The faecal microbiome in both species was dominated by the phyla *Firmicutes* and *Bacteroidetes*, typical of many vertebrates. Many phyla, including *Actinobacteria*, *Proteobacteria* and *Patescibacteria*, were shared between rabbits and hares. In contrast, bacteria from phylum *Verrucomicrobia* were present only in rabbits, while phyla *Lentisphaerae* and *Synergistetes* were represented only in hares. We did not identify phylum *Spirochaetes* in Australian hares; this phylum was previously shown to be present at high relative abundance in European hare faecal samples. These differences in the composition of faecal microbiota may be indicative of less discriminate foraging behaviour in rabbits, which in turn may enable them to adapt quicker to new environments, and may reflect the severe environmental impacts that this species has in Australia.

## INTRODUCTION

In a new environment, non-native species must face several barriers to first invade and then become established. They must quickly adapt to new environmental conditions while also competing with native species for food, shelter and water (*Čuda et al., 2015*;

Corresponding author
Robyn N. Hall, robyn.hall@csiro.au

*Shine, 2010*). European rabbits (*Oryctolagus cuniculus*) and European brown hares (*Lepus europaeus*) are lagomorphs in the family *Leporidae* and are both non-native species in Australia, originally introduced from Europe in 1859 (*Stott, 2003*). European rabbits rapidly colonised most of the Australian continent and are considered to be one of Australia's worst invasive vertebrate pests due to land degradation and competition with native animals and livestock (*Zenger, Richardson & Vachot-Griffin, 2003*; *Bird et al., 2012*; *Department of the Environment and Energy, 2016*). Despite foraging over wider areas, the impacts of European brown hares are less severe, although they are still considered to be a pest species (*Department of the Environment and Energy, 2016*; *Chapuis, 1990*). The specific factors underlying differences in colonising potential and impacts of each species are likely multifactorial, including differences in host physiology, reproductive strategies, behaviour, diet, and interaction with commensal and pathogenic microbes.

Recent advancements in sequencing technologies have highlighted the importance of interactions between the gut microbiome, host behaviour and physiology, and dietary preferences (*David et al., 2013*; *Henderson et al., 2015*; *Clemente et al., 2012*; *McKenna et al., 2008*; *Turnbaugh et al., 2007*; *Alcock, Maley & Aktipis, 2014*). Although the gastrointestinal microbiome of both rabbits and hares have previously been investigated, most studies to date were conducted on domestic rabbit populations in either Europe or China (*Eshar & Weese, 2014*; *Zeng et al., 2015*; *Kylie, Weese & Turner, 2018*; *Velasco-Galilea et al., 2018*; *Reddivari et al., 2017*; *Xing et al., 2019b*; *Dabbou et al., 2019*; *Xing et al., 2019a*; *O'Donnell et al., 2017*; *Crowley et al., 2017*; *Stalder et al., 2019*). Evaluating the microbial diversity in the gastrointestinal tract of wild hares and rabbits in Australia may provide additional insights as to how these two species differ, especially in terms of diet and/or metabolism (*Redford et al., 2012*). Furthermore, identifying differences between the gastrointestinal microbiota of lagomorphs in their native range compared to their introduced range may lead to novel insights into how to more sustainably manage introduced populations, for example by identifying pathogens that are absent in the introduced range that may be suitable for evaluation as biocontrol agents (*Redford et al., 2012*).

The microbiome can be investigated through a range of both traditional culture-dependent and less biased culture-independent techniques, including high throughput sequencing (*Gong et al., 2019*; *Wei et al., 2018*; *Jackson et al., 2013*; *Stefani et al., 2015*; *Öztürk et al., 2013*; *Adewumi et al., 2013*). Among various sequencing strategies, taxonomic profiling based on the 16S ribosomal RNA (rRNA) gene is widely used for estimating the bacterial and archaeal diversity in tissue or environmental samples due to the availability of well-curated databases such as SILVA (*Ranjan et al., 2016*). Broadly, high-throughput sequencing can be classified as either short-read sequencing, using the Illumina platform, or long-read sequencing, using platforms such as PacBio or Oxford Nanopore (*Schadt, Turner & Kasarskis, 2010*; *Lee et al., 2016*). Although all these platforms have previously been used to conduct 16S rRNA microbial profiling studies, direct comparisons of these methods are limited (*Shin et al., 2016*).

To investigate the differences between the gastrointestinal microbiome of Australian wild rabbits and hares we conducted 16S rRNA sequencing on faecal samples collected

from sympatric populations of these species using both Illumina and Oxford Nanopore sequencing platforms. The gastrointestinal microbial diversity of Australian lagomorphs was then compared to that observed in European populations to explore differences between populations in their native and invasive ranges.

# MATERIALS AND METHODS

## Sample collection

Faecal samples were collected at necropsy from nine wild European brown hares (*Lepus europaeus*) and 12 wild European rabbits (*Oryctolagus cuniculus*) of both sexes (Table S1). Although caecal contents may more accurately reflect the microbiota involved in digestion in these species, these samples were not collected for this study. The hares and rabbits used in this study were mostly adults or young adults, they were outwardly healthy, and were shot as part of routine vertebrate pest control operations in Mulligans Flat, ACT (−35.164, 149.165), between January and September 2016. Sample collection was approved by Barry Richardson, Oliver Orgill, and the ACT Parks and Conservation Service rangers of Mulligan's Flat Woodland Sanctuary. Mulligans Flat Nature Reserve comprises 781 hectares of box-gum grassy woodland with a sparse to moderately dense ground cover of grasses, herbs and shrubs. The reserve is fenced, preventing immigration and emigration of most animals. Rabbits and hares were shot from a vehicle using a 0.22-caliber rifle targeting the head or chest. Faecal pellets from the descending colon were collected during post-mortem and stored at −20 °C. All sampling was conducted according to the Australian Code for the Care and Use of Animals for Scientific Purposes as approved by CSIRO Wildlife and Large Animal Ethics Committee (approvals #12-15 and #16-02).

## Genomic DNA extraction

Genomic DNA was extracted from 50 mg of faecal pellets using the DNeasy Blood and Tissue kit as per manufacturer's instructions and described in detail at https://www.protocols.io/view/genomic-dna-extraction-from-animal-faecal-tissues-6bbhain (Qiagen, Chadstone Centre, Victoria, UK). Hare and rabbit samples were processed separately and reagent-only controls (ROC) were included with each set of extractions. All samples were quantified and evaluated for integrity using the Qubit dsDNA Broad Range assay kit and the Qubit Fluorometer v2.0 (Thermo Fisher Scientific, Waltham, MA, USA) and Nanodrop ND-1000 (Thermo Fisher Scientific, Waltham, MA, USA). Genomic DNA samples were stored at −20 °C.

## Illumina library preparation, sequencing and bioinformatics analysis

We amplified the V3–V4 region of the 16S rRNA gene (~460 bp) from genomic DNA and ROC according to the Illumina protocol for 16S Metagenomic Sequencing Library Preparation using a dual-indexing strategy, with modifications (*Illumina, 2013*). Briefly, initial PCR reactions (25 µl) were performed for each sample (including ROC) using 2× Platinum SuperFi PCR Master Mix (Thermo Fisher Scientific, Waltham, MA, USA), 0.5 µM of forward (5′-TCGTCGGCAGCGTCAGATGTGTATAAGAGACAGCCTAC

GGGNGGCWGCAG-3′) and reverse primer (5′-GTCTCGTGGGCTCGGAGATGTG TATAAGAGACAGGACTACHVGGGTATCTAATCC-3′) (overhang adapter sequence highlighted in bold) and 4.6 ng of genomic DNA as described at https://www.protocols.io/view/library-preparation-protocol-to-sequence-v3-v4-reg-6i7hchn. A 'no template control' (NTC) was included during reaction setup. Cycling conditions were: 98 °C for 30 s, followed by 25 cycles of 98 °C for 10 s, 55 °C for 15 s and 72 °C for 30 s, with a final extension of 10 min at 72 °C. PCR products were confirmed by agarose gel electrophoresis before being purified using AMPure XP beads (Beckman Coulter, Indianapolis, IN, USA) at a 1× ratio as described previously (*Illumina, 2013*). Each amplicon was then dual-indexed with unique DNA barcodes using the Nextera XT index kit (N7XX and S5XX, Illumina, San Diego, CA, USA) for PCR-based barcoding (*Illumina, 2013*). For each 50 µl PCR reaction, we used five µl of each index (i7 and i5), and five µl of the first PCR product. Cycling conditions were as described above but limited to eight cycles. Final libraries (including ROC and NTC) were purified with AMPure XP beads (Beckman Coulter, Indianapolis, IN, USA), validated using the Tapestation D1000 high sensitivity assay (Agilent Technologies, Santa Clara, CA) and the Qubit High Sensitivity dsDNA assay kit (Thermo Fisher Scientific, Waltham, MA, USA), pooled at equimolar concentrations, and sequenced on an Illumina MiSeq using 600-cycle v3 chemistry (300 bp paired-end) at CSIRO, Black Mountain, Canberra.

Illumina fastq reads were analysed in QIIME 2-2019.7 software (*Bolyen et al., 2019*). Raw fastq reads were quality filtered (i.e. filtered, dereplicated, denoised, merged and assessed for chimaeras) to produce amplicon sequence variants (ASV) using the DADA2 pipeline via QIIME2 (*Callahan et al., 2016*). The DADA2 generated feature table was filtered to remove ASVs at a frequency less than two, and remaining ASVs were aligned against the SILVA_132 (April 2018) reference database using QIIME2 feature-classifier BLAST+ (*Bokulich et al., 2018*; *Camacho et al., 2009*; *Quast et al., 2013*). Fasta sequences from DADA2 were also aligned to the same reference database using BLASTn (2.2.28) through the command line interface (CLI) using an *e*-value cut-off of 1E−90 (Table S2). We processed and exported BLASTn outputs into QIIME2 to perform taxonomic analysis. All scripts used were deposited at https://github.com/SomaAnand/Hare_rabbit_microbiome. Raw sequence data were deposited in the sequence read archive of NCBI under accession number PRJNA576096.

The microbial diversity and richness in individual samples was estimated using alpha diversity metrics (Shannon index and observed OTU), and the variation in overall composition between rabbits and hares was measured using beta diversity metrics (Bray–Curtis dissimilarity) after rarefaction at a subsampling depth of 149,523 using the q2-diversity pipeline within QIIME2 (*Faith, Minchin & Belbin, 1987*; *Anderson, 2001*). This subsampling depth was chosen based on the sample (excluding one ROC (rabbit) and PCR NTC) with the lowest number of reads.

**Nanopore library preparation, sequencing and bioinformatics analysis**
The entire 16S rRNA gene (~1,500 bp) was targeted for sequencing using the Nanopore MinION sequencing platform. We amplified the 16S rRNA gene from all genomic DNA

samples using the universal primers 27F and 1492R (*Weisburg et al., 1991*). PCR reactions (100 µl) were performed for each sample using 2× Platinum SuperFi PCR master mix (Thermo Fisher Scientific, Waltham, MA, USA), 0.4 µM each primer, and 11.5 ng genomic DNA as described at https://www.protocols.io/view/library-preparation-protocol-to-sequence-full-leng-6j6hcre. Cycling conditions were: 98 °C for 30 s, followed by 28 cycles of 98 °C for 10 s, 55 °C for 15 s and 72 °C for 40 s, with a final extension of 5 min at 72 °C. PCR products were confirmed by agarose gel electrophoresis before being purified with AMPure XP beads (Beckman Coulter, Indianapolis, IN, USA), validated using the Tapestation D1000 high sensitivity assay (Agilent Technologies, Santa Clara, CA, USA), and quantified using the Qubit High Sensitivity dsDNA assay kit (Thermo Fisher Scientific, Waltham, MA, USA). For library preparation we used the ligation sequencing kit 1D (SQK-LSK108) in combination with the native barcoding kit 1D (EXP-NBD103) (Oxford Nanopore Technologies, Oxford, UK) as per the manufacturer's protocol, except 500 ng of input DNA per sample was used for end preparation (*Hu & Schwessinger, 2018*). For the addition of barcodes, 80 ng of end-prepped DNA was used as input, and barcoded samples were pooled in equimolar concentrations to obtain at least 400 ng of pooled DNA. We used a final library amount of 200 ng to obtain maximum pore occupancy on a MinION R.9.4.1 flow cell. Two MinION flowcells were used to run all samples and each flow cell had a mix of hare and rabbit samples to control for potential batch effects.

Nanopore raw reads in fast5 format were demultiplexed using deepbinner (*Wick, Judd & Holt, 2018*). Basecalling, adapter trimming, and conversion into fastq format was performed in Guppy 2.3.7 (Oxford Nanopore Technologies, Oxford, UK). BLASTn was used to locally align 16S sequences with a quality score higher than seven against the SILVA_132 reference database. Top hits were exported in Biological Observation Matrix format and imported into QIIME2 for subsequent analyses. Nanopore data was rarefied at a subsampling depth of 109,435 for diversity analysis based on the sample with the fewest Nanopore reads, in order to mimic the Illumina workflow. This rarefied dataset was then used to analyse the taxonomic results as per the Illumina workflow described above. Species level taxonomic classification was performed through the EPI2ME platform (Oxford Nanopore, Oxford, UK) (Table S2). Raw sequence data were deposited in the sequence read archive (SRA) of NCBI under accession number PRJNA576096.

## Statistical analysis

We assessed whether the bacterial diversity of hares and rabbits was statistically different by using permutation-based statistical testing (PERMANOVA) via the QIIME diversity beta-group-significance pipeline within QIIME2 for both Illumina and Nanopore datasets (*Anderson, 2001*). Statistical significance in alpha diversity was estimated using the nonparametric Kruskal–Wallis test within the R package 'ggpubr' (*Kruskal & Wallis, 1952*). We also evaluated statistical differences between hare and rabbit faecal samples for each observed bacterial phyla using a combination of multiple tests. We performed analysis of variance (ANOVA) to estimate the variance of both populations and Student's $t$-test to identify statistical differences between the means of both groups. Estimated
*p*-values were corrected using the Benjamini–Hochberg method to control the False Discovery Rate for multiple hypothesis testing. Pearson correlation coefficients were calculated to assess the correlation between Illumina and Nanopore sequencing results using the scipy.stats package in Python.

# RESULTS

## Interrogating hare and rabbit faecal microbiota using 16S rRNA sequencing

We performed Illumina short-read sequencing of the 16S rRNA V3–V4 region and Nanopore long read sequencing of the entire 16S rRNA gene on faecal samples of wild hares and rabbits living sympatrically in a periurban nature reserve in Australia. Short-read sequencing of 24 samples through the Illumina MiSeq pipeline produced an aggregate of 14,559,822 reads greater than Q30 (mean 501,908 reads per sample, excluding ROC and NTC). Reads were converted to ASVs, which were assigned using the SILVA_132 reference database via BLAST+ in QIIME2. This produced 6,662 unique features at a frequency greater than two. Long-read sequencing of 21 samples (no ROCs or NTC due to lack of amplification) using the Nanopore MinION platform produced an aggregate of 6,544,770 reads (mean of 311,656 reads per sample) above a quality threshold of seven across two MinION runs. After aligning the reads against the SILVA_132 reference database using BLAST, 47,100 unique features with a frequency greater two were obtained.

## The faecal microbiome composition varies significantly between sympatric wild rabbits and hares, and between individual rabbits

We conducted alpha and beta diversity analyses to assess the species richness, evenness, and composition both within individual animals (alpha diversity) and between rabbits and hares (beta diversity). The richness and evenness of bacterial species in individual animals was not significantly different between rabbits and hares, as measured by the Shannon index and Kruskal–Wallis test (Fig. S1). However, the beta diversity between rabbit and hare faecal samples differed significantly ($p = 0.01$ as measured by Bray–Curtis dissimilarity matrix and PERMANOVA) (Fig. 1). We also observed considerable variance in faecal microbial beta diversity between individual rabbits (Fig. 1). In contrast, the hare faecal samples examined all had a similar bacterial composition with relatively low variance between individuals. We observed these effects in both the Illumina and Nanopore 16S sequencing data sets (Fig. 1). We did not observe clear correlation between bacterial diversity and sex, reproductive status, or season of sample collection, although our statistical power was low due to the small sample size. Although there were more pregnant and/or lactating rabbits than hares, which may lead to biases in the observed bacterial diversity, this difference was not statistically significant (Fisher's exact test, $p = 0.14$). Similarly, the variances of bodyweight for each population were not significantly different (*F*-test, $p = 0.14$).

Taxonomic classification of both the Illumina and Nanopore sequencing data sets identified *Firmicutes* and, to a lesser extent *Bacteroidetes*, to be the two dominant phyla in

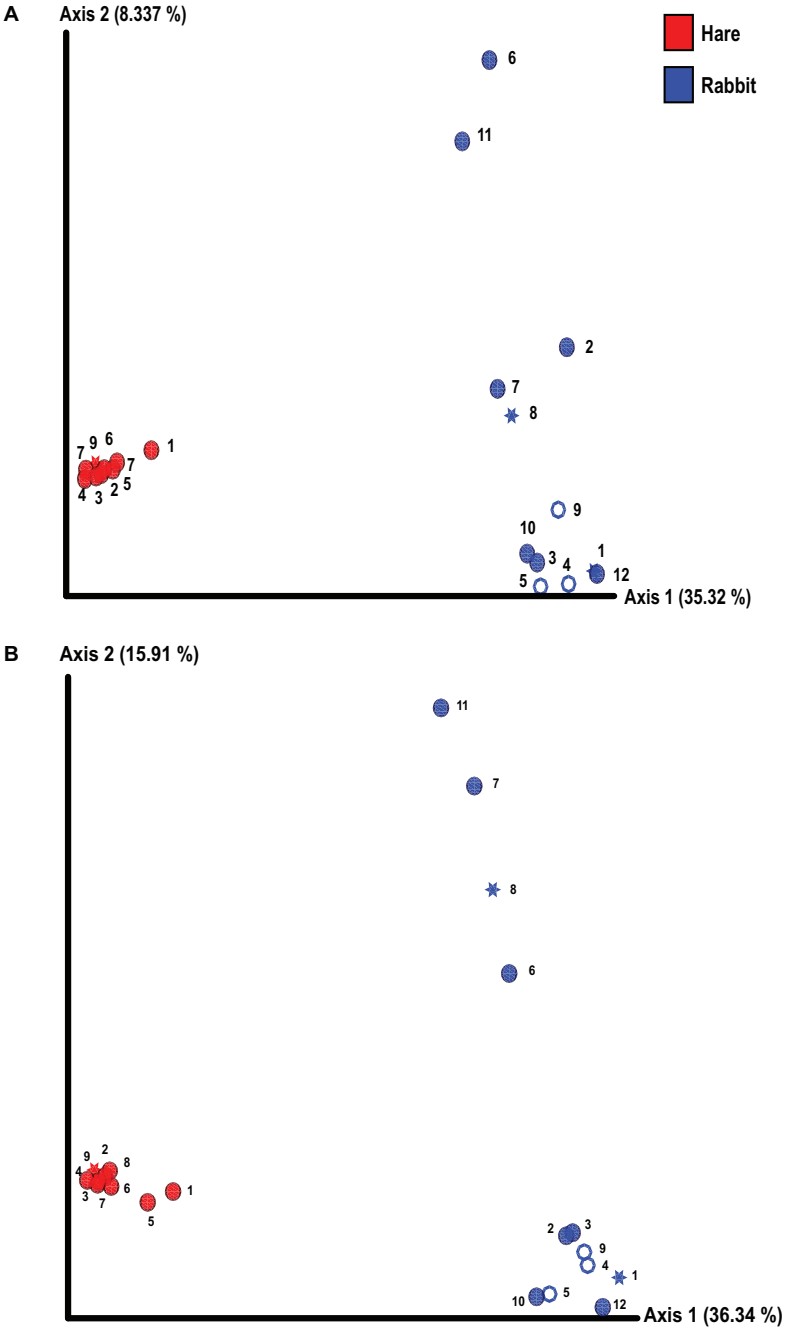

**Figure 1 The composition of faecal microbial communities varied significantly between wild rabbits and hares and between individual rabbits.** 16S rRNA profiling was conducted on wild hare and rabbit faecal samples using either (A) Illumina, or (B) Nanopore sequencing platforms. Principal coordinates analysis of beta diversity using Bray-Curtis dissimilarity (as calculated in QIIME2) was used to compare the microbial species compositions. Significance was assessed using PERMANOVA. Numbers refer to individual animal identifiers as described in Table S1. Stars symbolises pregnant and lactating females, open circles symbolise lactating females, and closed circles symbolise non-pregnant and non-lactating animals.

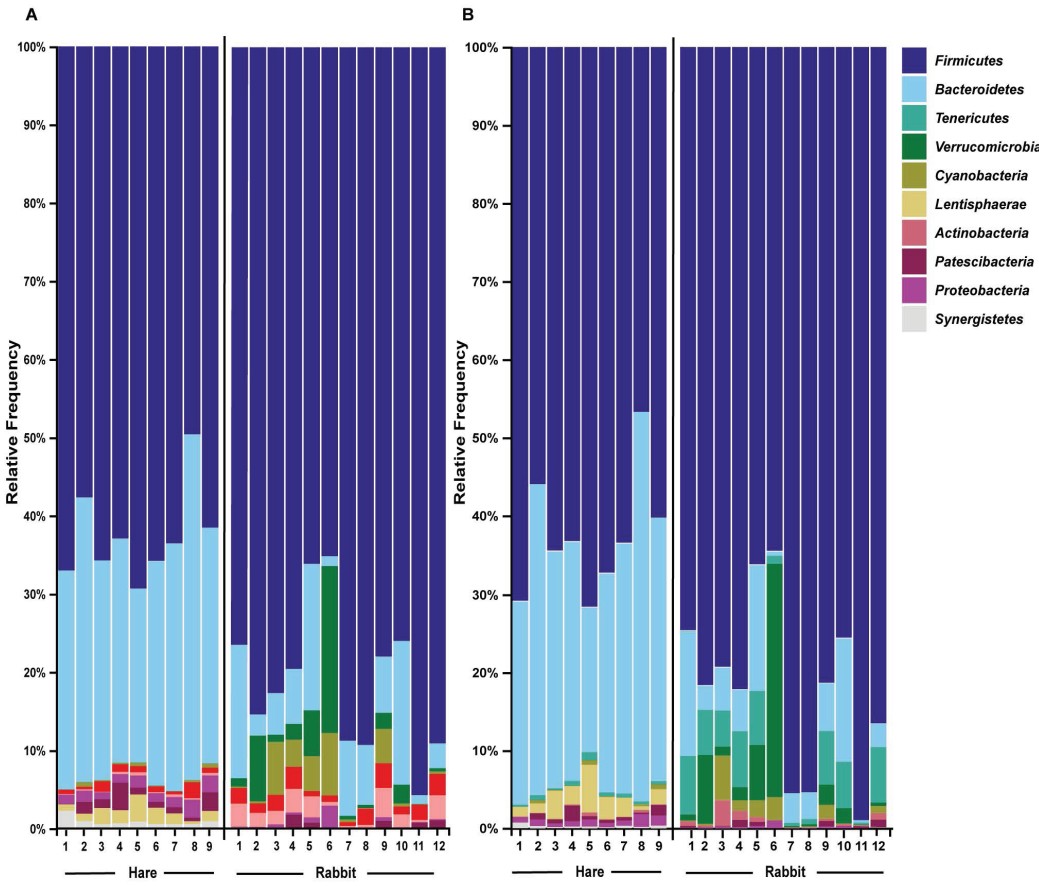

**Figure 2 The faecal microbiomes of wild rabbits and hares were distinct at phylum level.** Taxonomic classification of 16S rRNA sequences from (A) Illumina and (B) Nanopore sequencing platforms was performed by alignment to the SILVA_132 reference database using BLAST+ within QIIME2 for Illumina data or BLASTn for Nanopore data. The relative frequency of reads assigned to bacterial phyla present at an abundance greater than 0.5% were plotted for each sample.

both hare and rabbit faecal samples (Fig. 2). On average, *Firmicutes* and *Bacteroidetes* together comprised more than 85% of the faecal microbiome in hares and rabbits and across both platforms. Hare faecal samples contained a significantly higher ratio of *Firmicutes* to *Bacteroidetes* compared to rabbit faecal samples ($p = 0.015$ as measured by Student's *t*-test). In both hare and rabbit faecal samples, *Firmicutes* were predominantly represented by the families *Ruminococcaceae*, *Christensenellaceae*, *Lachnospiraceae* (genus *Roseburia*), *Eubacteriaceae* and *Erysipelotrichaceaea*, while *Bacteroidetes* were predominantly represented by the families *Rikenellaceae* and *Barnesiellaceae* (Fig. 3; Fig. S2). Within the *Firmicutes* and *Bacteroidetes*, the families *Marinifilaceae* (genera *Odoribacter* and *Butryicimonas*), *Tannerellaceae* (genus *Parabacteroides*) and *Veillonellaceae* were more abundant in hares compared to rabbits, while the families *Clostridiaceae 1*, *Enterococcaceae*, *Planococcaceae*, *Lactobacillaceae*, *Peptostreptococcaceae* and *Bacillaceae* were more abundant in rabbits, although not all families were present in all rabbits (Fig. 3; Fig. S2). Phylum *Tenericutes* (genus *Anaeroplasmataceae*) was present

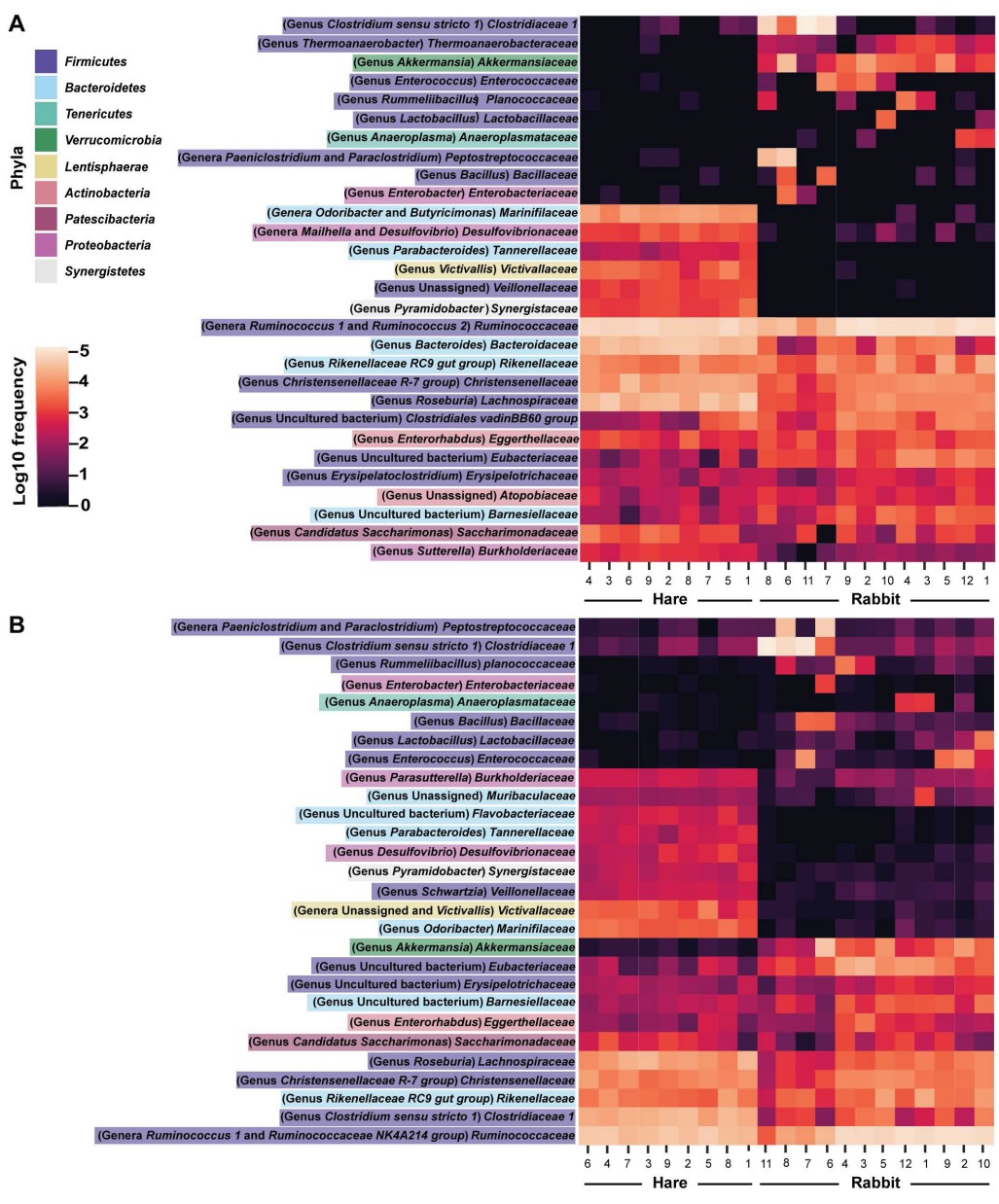

**Figure 3** **Faecal microbial diversity of Australian wild hares and rabbits at the family level identified unique profiles for wild hares and rabbits.** Taxonomic classification of 16S rRNA sequences from (A) Illumina and (B) Nanopore sequencing platforms was performed by alignment to the SILVA_132 reference database using BLAST+ within QIIME2 for Illumina data or BLASTn for Nanopore data. The frequency of reads mapping to bacterial families for individual samples are shown in a heatmap on a log10 scale. Family and genera names are highlighted according to phylum. Rows are clustered based on features identified across all samples. Bacterial families present at a frequency less than 0.5% are not included.

at a significantly higher relative frequency in rabbit faecal samples ($p < 0.01$ as measured by Student's $t$-test). Phylum *Verrucomicrobia* (genus *Akkermansia*) was identified only in rabbit samples and not in hare samples, while phyla *Lentisphaerae* (genus *Victivallis*) and

**Table 1** Top ten bacterial species identified in hare and rabbit faecal samples using Nanopore sequencing.

| Phylum | Family | Genus | Species | Mean relative abundance (%) |
|---|---|---|---|---|
| Hares | | | | |
| Firmicutes | Ruminococcaceae | Ruminococcus | albus | 11.360 |
| Bacteroidetes | Bacteroidaceae | Bacteroides | uniformis | 3.712 |
| Firmicutes | Ruminococcaceae | Ruminococcus | champanellensis | 3.251 |
| Bacteroidetes | Bacteroidaceae | Bacteroides | sartorii | 1.578 |
| Firmicutes | Ruminococcaceae | Ruminococcus | flavefaciens | 1.546 |
| Bacteroidetes | Bacteroidaceae | Bacteroides | mediterraneensis | 1.471 |
| Bacteroidetes | Bacteroidaceae | Bacteroides | eggerthii | 1.322 |
| Bacteroidetes | Bacteroidaceae | Bacteroides | caecigallinarum | 1.068 |
| Bacteroidetes | Rikenellaceae | Alistipes | putredinis | 1.066 |
| Firmicutes | Lachnospiraceae | Roseburia | hominis | 0.980 |
| Rabbits | | | | |
| Firmicutes | Clostridiaceae | Clostridium | baratii | 4.239 |
| Firmicutes | Clostridiaceae | Clostridium | moniliforme | 2.837 |
| Firmicutes | Peptostreptococcaceae | Paeniclostridium | sordellii | 2.258 |
| Firmicutes | Clostridiaceae | Clostridium | paraputrificum | 1.710 |
| Firmicutes | Peptostreptococcaceae | Paraclostridium | bifermentans | 1.619 |
| Firmicutes | Clostridiaceae | Clostridium | senegalense | 1.541 |
| Firmicutes | Ruminococcacaea | Ruminococcus | albus | 1.520 |
| Firmicutes | Clostridiaceae | Clostridium | argentinense | 0.873 |
| Firmicutes | Peptostreptococcaceae | Paraclostridium | benzoelyticum | 0.764 |
| Verrucomicrobia | Akkermansiaceae | Akkermansia | glycaniphila | 0.728 |

*Synergistetes* (genus *Pyramidobacter*) were identified only in hare samples. Other phyla such as *Proteobacteria* (genus *Sutterella* in both hares and rabbits, family *Desulfovibrionaceae* in hares only, and genus *Enterobacter* in rabbits only), *Actinobacteria* (genus *Eggerthellaceae*) and *Patescibacteria* (family *Saccharimonadaceae*) were present at low abundance (0.5–3%) in both rabbit and hare samples (Figs. 2 and 3; Fig. S2).

We were also able to perform taxonomic classification of Nanopore sequencing data to the species level, due to the longer read lengths associated with this sequencing technology. There was considerable variation in bacterial species between individual faecal samples and many reads remained unassigned at the species level. However, *Ruminococcus albus* clearly dominated the hare gastrointestinal microbiome, being present in all hare faecal samples at a mean relative frequency of 11.4% (Table 1). This bacterial species was also consistently detected in rabbit faeces, but at a much lower relative frequency (1.5%). Of the ten most abundant individual species, hare faeces were dominated by *Ruminococcus* sp, *Bacteroides* sp, *Alistipes putredinis*, and *Roseburia hominis*; in contrast, rabbit faeces were dominated by several *Clostridium* and *Paraclostridium* species, as well as *Paeniclostridium sordellii*, *Ruminococcus albus*, and *Akkermansia glycaniphila* (Table 1).

The NTC microbiome comprised the phyla *Proteobacteria* and *Bacteroidetes* at relative frequencies of 99% and 1%, respectively, from a total of 2,158 raw reads (Fig. S3). When analysing the ROCs, there was obvious genomic DNA contamination of both ROCs, with several phyla shared between faecal samples and ROCs (Fig. S3). Additional bacterial phyla were also detected only in ROCs, likely comprising the 'reagent microbiome,' including *Plantomycetes*, *Dependentiae*, *Chloroflexi* and *Chlamydiae*. The total number of raw reads in the rabbit ROC was 6,279 (compared to the mean 501,908 reads/sample), while contamination was more pronounced in the hare ROC, which produced 265,118 raw reads. Since the phyla-level distribution of reads in the hare ROC strongly reflected that seen in all hare samples, it was assumed that this contamination would not grossly alter the results, particularly because we were not dealing with low-biomass samples.

### Geography alters the faecal microbiome of wild hares

We then compared the faecal microbiome of Australian wild hares to that of wild hares in their native range of Europe (sampled and sequenced in a previous study (*Stalder et al., 2019*)) to investigate the influence of geography on the gastrointestinal microbiome. Strikingly, phylum *Spirochaetes* was the third most abundant phylum in faecal samples of European origin (Fig. 4), yet it was only present at very low abundance from Australian hares and rabbits. Furthermore, phylum *Patescibacteria* was below our limit of detection (<0.5% relative abundance) in European hare samples, while in Australian hare samples it was present at a higher relative abundance (1.3%). Other phyla were detected in both European and Australian hare samples at similar relative abundances (Figs. 2 and 4).

### Nanopore and Illumina sequencing platforms reveal a similar faecal microbiome independent of platform

We found a strong correlation (as measured by Pearson correlation coefficient) between Illumina and Nanopore sequencing results at the phylum ($r = 0.945$, $p < 0.001$), family ($r = 0.947$, $p < 0.001$) and genus ($r = 0.923$, $p = <0.001$) levels (Figs. 2 and 3; Fig. S2). Most notably, phyla *Actinobacteria* and *Synergistetes* were present at a higher relative abundance in the Illumina dataset, while phyla *Tenericutes*, *Lentisphaerae*, and to a lesser extent *Cyanobacteria*, were present at higher abundance in the Nanopore dataset (Fig. 2). At the family level, *Thermoanaerobacteraceae*, *Bacteroidaceae*, *Clostridiales vadinBB60* group and *Atopobiaceae* were detected in the Illumina but not the Nanopore data, while *Muribaculaceae* and *Flavobacteriaceae* were identified only in the Nanopore data (Fig. 3; Fig. S2). Even at the genus level there was generally good agreement between datasets (Fig. 3). However, genera *Butyricimonas*, *Mailhella*, *Ruminococcus 2*, *Erysipelatoclostridium* and *Sutterella* were only detected in Illumina data and *Parasutterella*, *Schwartzia*, an unassigned *Victivallaceae*, and the *Ruminococcaceae NK4A214 group* were only identified in the Nanopore data.

## DISCUSSION

To date, studies investigating the gastrointestinal microbial diversity of lagomorphs have been largely limited to domestic production rabbits in either Europe or China

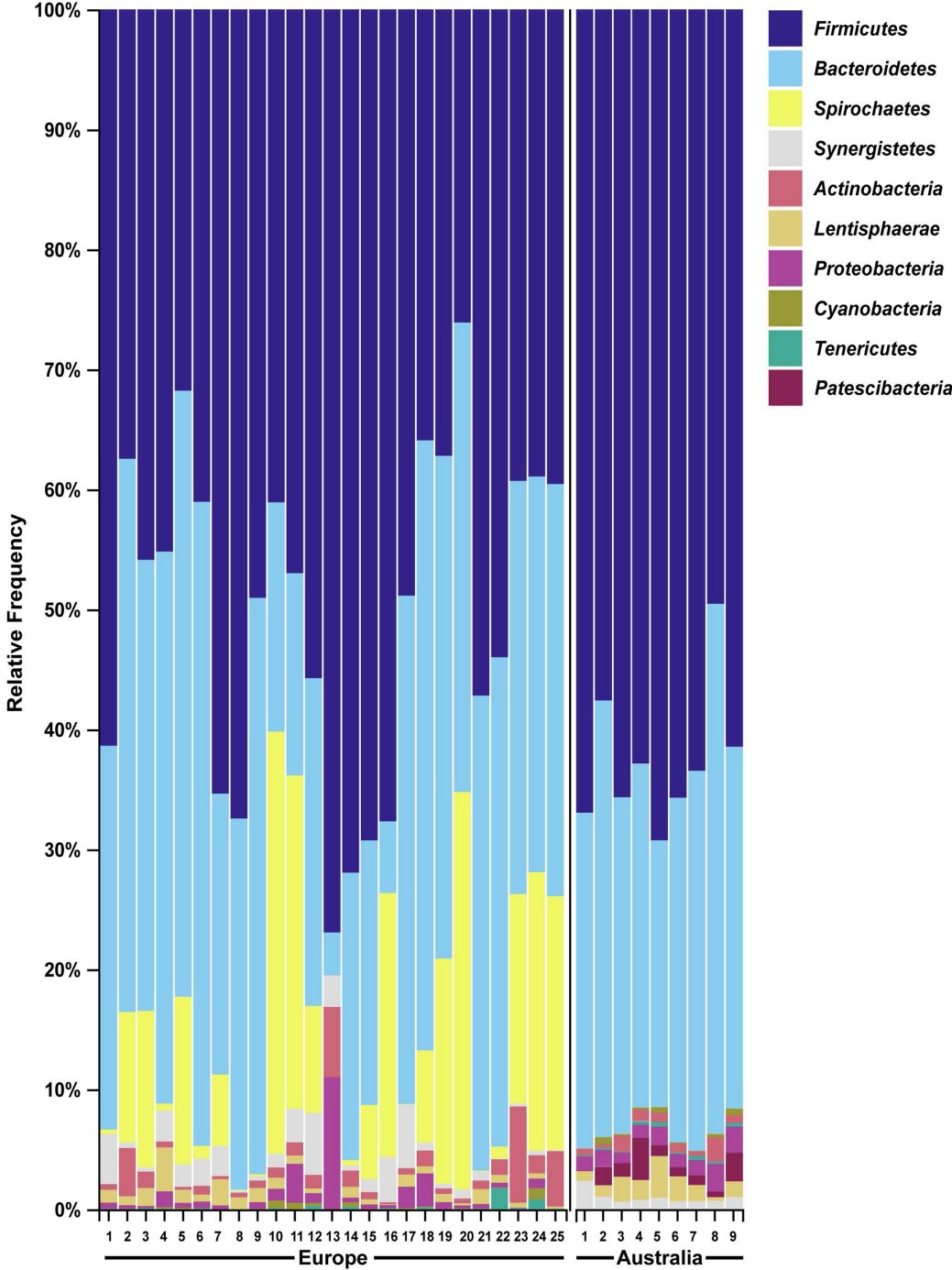

**Figure 4** **Faecal microbial diversity of wild hares in their native range of Europe shows *Spirochaetes* to be the third most abundant bacterial phyla.** Raw Illumina sequencing data were obtained from a previous study examining the 16S rRNA diversity in faecal pellets of European brown hares in Europe (*Stalder et al., 2019*). Reads were processed and classified in parallel with sequencing data from Australian hare faecal samples, using BLAST+ as implemented in QIIME2 against the SILVA_132 reference database. The relative frequency of reads assigned to bacterial phyla present at an abundance greater than 0.5% were plotted for each sample. Australian hare data presented here are repeated from Fig. 2 for ease of comparison.                                 

(*Eshar & Weese, 2014*; *Zeng et al., 2015*; *Kylie, Weese & Turner, 2018*; *Velasco-Galilea et al., 2018*; *Xing et al., 2019b*). Currently, two studies have focussed on wild lagomorphs; the first investigating wild rabbits in Wales (*Crowley et al., 2017*) and the second examining European brown hares in their native mainland European home-range in Germany and Austria (*Stalder et al., 2019*). We are interested in understanding why rabbits, an invasive pest species in Australia, were able to rapidly colonise over two thirds of the continent, an area 25 times the size of Britain, within 50 years. In contrast, populations of European brown hares, also a non-native species, remained relatively stable, despite both species occupying a similar dietary niche. Differences in colonising potential of these two species are likely multifactorial, for example involving differences in behaviour, physiology and dietary preferences. Comparative analysis of the gastrointestinal microbiome of these two species is one method that can be used to investigate host differences, which may in turn reveal clues to the differences in diet and potentially environmental impacts that each species has. A secondary aim was to use this study as an opportunity to compare short-read (Illumina) and long-read (Nanopore) sequencing technologies for microbial diversity investigations.

We estimated the gastrointestinal microbiome from faecal samples collected from nine wild European brown hares and twelve European rabbits living sympatrically in their non-native range in Australia. Rabbits had significantly different faecal microbiome compositions compared to hares and showed greater variability in bacterial composition between individuals. Species richness is crucial for imparting resilience to microbial communities, facilitating rapid and effective adaptation to new environmental conditions (*Lozupone et al., 2012*). Species-rich communities can better resist invading pathogens and decreased microbial diversity in humans has been linked to a range of pathologies (*Lozupone et al., 2012*). Although individual hares and rabbits had similar species richness in their faecal microbiomes, at a population level, we found that rabbits varied widely in their species compositions compared to hares. It is interesting to speculate that this may allow for subsets of the population to rapidly expand into new environments. Whether this is an intrinsic feature of *Oryctolagus cuniculus* or whether this is due to their larger population remains to be determined. In contrast, hares may face more of a 'microbiome bottleneck,' which may restrict their ability to adapt to new environments or diets. Hares are known to be more selective foragers than rabbits, which are considered more generalist (*Chapuis, 1990*). The observed diversity in the faecal microbiome composition of wild rabbits may be a consequence of highly variable individual dietary preferences, or alternatively, a more diverse faecal microbiome may indeed permit rabbits to consume feeds that hares cannot digest.

The faecal microbiomes of both host species were dominated by the phyla *Firmicutes* and *Bacteroidetes*, as is typical for other vertebrate species (*Eckburg et al., 2005*). However, the ratio of *Firmicutes* to *Bacteroidetes* varied, being markedly higher in rabbits compared to hares. At the genus level, hare faeces were dominated by *Ruminococcus* and *Bacteroides* species, while rabbit faeces were dominated by *Clostridium* and *Paraclostridium* species. In humans, an increased *Firmicutes* to *Bacteroidetes* ratio has notoriously been associated with age, diet and obesity (*Ley et al., 2006*). A higher

abundance of *Bacteroidetes* (genus *Bacteroides*) has been linked to consumption of diets rich in protein and fat in humans (*Wu et al., 2011*), and indeed, hares are known to prefer diets rich in crude fat and protein (*Schai-Braun et al., 2015*). Despite having a higher daily digestible nitrogen intake, hares tend to have less efficient protein digestion compared to rabbits, potentially due to the absence of key microbes in their gastrointestinal tract (*Kuijper, Van Wieren & Bakker, 2004*). Another apparent difference in the faecal microbiomes of these species was the presence of phylum *Verrumicrobia* in rabbits. This phylum is dominated by bacteria from the genus *Akkermansia*, which are noted to be positively influenced by dietary polyphenols (*Roopchand et al., 2015*; *Anhê et al., 2016*). Polyphenols are metabolised by intestinal bacteria to generate short-chain fatty acids (SCFAs), and previous studies have demonstrated that rabbits produce a higher concentration of SCFAs than hares, with a higher ratio of butyrate to propionate (*Miśta et al., 2015*; *Marounek, Brezina & Baran, 2000*; *Miśta et al., 2018*). Again, it is unknown whether the *Akkermansia* detected in rabbits in this study permit the digestion of these polyphenols, influencing the dietary preference of rabbits, or whether the intake of polyphenols supports a detectable *Akkermansia* population in rabbits.

Phyla *Lentisphaerae* and *Synergistetes* were observed only in hare populations and were dominated by the genera *Victivallis* and *Pyramidobacter*, respectively. Age and diet appear to be important factors in regulating the abundance of these phyla (*Niu et al., 2015*; *Amato et al., 2015*). A previous study in pigs reported a decrease in relative abundance of phyla *Lentisphaerae* and *Synergistetes* with ageing (*Niu et al., 2015*). Furthermore, lower relative abundances of phyla *Proteobacteria*, *Lentisphaerae* and *Tenericutes* in vervets and humans have been associated with 'Western diets' rich in carbohydrates (*Amato et al., 2015*). The observed differences in faecal microbiota could also be related to other known differences in digestive physiology between rabbits and hares. For example, hares have a higher gastrointestinal passage rate compared to rabbits, while rabbits retain digesta longer in order to maximise the efficiency of nutrient extraction (*Kuijper, Van Wieren & Bakker, 2004*). Rabbits also have a greater ability to digest hemicelluloses and have a higher rate of methanogenesis compared to hares (*Miśta et al., 2015*; *Marounek, Brezina & Baran, 2000*; *Miśta et al., 2018*; *Stott, 2008*).

We also analysed raw data sequenced in a previous study examining faecal pellets of wild hares in their native range of Europe to estimate the influence of geography on the gut microbiome (*Stalder et al., 2019*). We observed the phylum *Spirochaetes* to be the third dominant phyla in the European dataset, in accordance with the original analysis. However, in our Australian dataset only very few reads were associated with this phylum (Table S2). Although bacteria within this phylum can be pathogenic, the *Spirochaetes* identified in European hares were associated with a non-pathogenic genus. The absence of this phylum of bacteria in Australian hares may reflect geographical differences in diet between the populations studied, or loss of this bacterial species after introduction into Australia. For example, the hares sampled in this study had access to tropical and native Australian grasses such as *Panicum effusum*, *Themeda australis*, *Eragrostis benthamii*, *Stipa bigeniculata* and *Austrostipa scabra*, which would not have been available to the hares
sampled in the European study. In contrast, hares in Europe have been reported to preferentially graze on cereal crops and grasses such as soybean, beet plants, barley, alfalfa, knotgrass, chickweed, clover and maize (*Schai-Braun et al., 2015*). These species were not available to the hares sampled in our Autralian study. Alternatively, different sample handling, for example, different primers used for 16S amplification, may possibly have resulted in selective bias against *Spirochaetes* in our study. Despite these spirochaetes likely being non-pathogenic, it is worth noting that the absence of pathogens of invasive species in their non-native range may contribute to spread and persistence of these alien species in new environments. Since faecal samples in this study were collected from healthy hares and rabbits, no candidate pathogens were investigated here.

An additional aim of this study was to compare the bacterial diversity between two widely used sequencing platforms. The Illumina platform is a popular approach for 16S rRNA sequencing, particularly because of its lower error rate (*Pollock et al., 2018*). However, the short-read length can make species level identification very challenging, especially between closely related species (*Pollock et al., 2018*). In contrast, the Oxford Nanopore platform has the ability to sequence very long reads, however, it is prone to a relatively high error rate, again making accurate taxonomic assignment challenging (*Pollock et al., 2018*). In this study, we observed very similar bacterial diversity at the phylum, family, and genus levels with both sequencing platforms, confirming the suitability of either technology for 16S rRNA studies and further confirming the biological relevance of our findings. The minor observed differences in relative abundances of different phyla are most likely due to PCR-based errors or bias, since different primer sets were used for each sequencing platform, or bias during sequencing (*Pollock et al., 2018*). The strong correlation between Illumina and Nanopore data in this study suggests that despite the higher error rate of Nanopore technology, this is not a significant issue for taxonomic classification, at least at the genus level or higher. With the additional benefits of longer reads and species level classification of Nanopore data, this study highlights that there are clear benefits to the use of the Nanopore platform for metagenomics studies.

## CONCLUSIONS

In conclusion, we observed notable differences in the microbiome of hares and rabbits living in sympatry in their non-native range in Australia. Though these species inhabit the same habitat, their behaviour and dietary preferences clearly influence differences in faecal bacterial composition. The more variable gastrointestinal microbiota of rabbits compared to hares could be a contributing factor in their ability to spread very successfully and establish in new environments. Additionally, the absence of bacteria from the phylum *Spirochaetes* in Australian compared to European wild hares (where it was observed frequently and often at high relative abundance) demonstrates considerable geographical differences between populations, although whether these spirochaetes are beneficial or detrimental to hares was not determined. Future studies correlating different

bacterial species in lagomorph microbiomes with specific plant species may provide further insights into the impacts of wild rabbits and hares in Australia.

## ACKNOWLEDGEMENTS

We thank Barry Richardson, Oliver Orgill and the ACT Parks and Conservation Service rangers of Mulligan's Flat Woodland Sanctuary for their assistance with sampling. We also thank Gabrielle Stalder, Stefanie Wetzels, and Evelyne Mann for sharing their previously published raw data from hares in Europe. We also wish to thank Amanda Padovan and Maria Jenckel for critical revision of the manuscript.

### Funding

Funding was provided by a Centre for Biodiversity Analysis 'Ignition' Grant, a collaborative initiative of the Australian National University, CSIRO and the University of Canberra. The funders had no role in study design, data collection and analysis, decision to publish, or preparation of the manuscript.

### Grant Disclosures

The following grant information was disclosed by the authors:
Centre for Biodiversity Analysis 'Ignition' Grant.
Australian National University, CSIRO, University of Canberra.

### Competing Interests

Tanja Strive and Robyn N. Hall receive research funding for wild rabbit management via the Centre for Invasive Species Solutions, a national collaborative research, development and extension organisation co-funded by government, research, and agricultural industry sectors.

### Author Contributions

- Somasundhari Shanmuganandam conceived and designed the experiments, performed the experiments, analysed the data, prepared figures and/or tables, authored or reviewed drafts of the paper, and approved the final draft.
- Yiheng Hu performed the experiments, analysed the data, authored or reviewed drafts of the paper, and approved the final draft.
- Tanja Strive conceived and designed the experiments, authored or reviewed drafts of the paper, contributed to sample collection, and approved the final draft.
- Benjamin Schwessinger conceived and designed the experiments, analysed the data, prepared figures and/or tables, authored or reviewed drafts of the paper, and approved the final draft.
- Robyn N. Hall conceived and designed the experiments, analysed the data, prepared figures and/or tables, authored or reviewed drafts of the paper, and approved the final draft.
## Animal Ethics

The following information was supplied relating to ethical approvals (i.e. approving body and any reference numbers):

All sampling was conducted according to the Australian Code for the Care and Use of Animals for Scientific Purposes as approved by CSIRO Wildlife and Large Animal Ethics Committee (approvals #12-15 and #16-02).

## Field Study Permissions

The following information was supplied relating to field study approvals (i.e. approving body and any reference numbers):

Sample collection was approved by Barry Richardson (ACT Scientific Committee), Oliver Orgill (Invasive Animals manager, ACT Parks and Conservation Service), and the ACT Parks and Conservation Service rangers of Mulligan's Flat Woodland Sanctuary.

## Data Availability

Raw Illumina sequence data and Nanopore data are available in the SRA at NCBI: PRJNA576096.

Detailed protocols are available at https://www.protocols.io/researchers/somasundhari-shanmuganandam/publications. All scripts used are available at GitHub: https://github.com/SomaAnand/Hare_rabbit_microbiome.

## Supplemental Information

Supplemental information for this article can be found online at http://dx.doi.org/10.7717/peerj.9564#supplemental-information.

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
