# Peer review of "Uncovering the microbiome of invasive sympatric European brown hares and European rabbits in Australia"

_PeerJ, doi:10.7717/peerj.9564_

## Round 0.1 · original submission · Minor Revisions

· Academic Editor

Minor Revisions

Please provide a point-by-point response to all of the reviewers' comments.

·

Basic reporting

This is a very interesting paper. I have flagged up a couple of anomalies in terms of what has been reported within it from existing literature, but these should be relatively easy to address.

Experimental design

Generally a good and solid approach. As I have mentioned, I am rather surprised that the authors did not take the opportunity to analyse the caecal contents in this work, but nevertheless the data from faecal samples is novel and interesting.

Validity of the findings

One question about figure 5 but otherwise no issues.

Additional comments

A few typos which could dramatically change the course of the content. Otherwise nothing to add.

Reviewer 2 ·

Basic reporting

'no comment'

Experimental design

'no comment'

Validity of the findings

I only suggest to improve discussion in following lines:
In lines (407-408), authors should improve the evidence to support the idea that Nanopore platform is the technology of choice for metagenomics studies, because I think before that, they should correct nanopore reads before the taxonomic assignation, to make that statement.
In lines 385-386. I suggest that authors should add the geographic difference and the kind of plants that they have in the area to to support the idea of "absence Spirochetes in hares is a reflection of geographical differences".

Additional comments

Manuscript is well writing and it has a novel informations, but I suggest to improve discussion in following lines:
In lines (407-408), authors should improve the evidence to support the idea that Nanopore platform is the technology of choice for metagenomics studies, because I think before that, they should correct nanopore reads before the taxonomic assignation, to make that statement.
In lines 385-386. I suggest that authors should add the geographic difference and the kind of plants that they have in the area to to support the idea of "absence Spirochetes in hares is a reflection of geographical differences".

·

Basic reporting

no comment

Experimental design

no comment

Validity of the findings

no comment

Additional comments

The manuscript submitted by Shanmuganandam et al studied microbial community structure of European hares and rabbits in Australia. The authors found significant difference in microbial composition between hare and rabbit in terms of beta diversity and taxonomic structure. Also, authors compared microbiome in hares in Australia and those in Europe and found some differences in taxonomy, especially absence and presence of the phylum Spirochaetes. They also stated the similarity of community structure of hares/rabbits fecal microbiome analyzed by difference sequencing platforms, Illumina and Nanopore.
Generally, this experiment was precisely conducted and clearly documented. I would recommend it for acceptance after a few minor concerns listed below are addressed.

L152, In this study, the raw sequence reads were converted to ASVs. Authors should clarify the cut-off similarity for "OTU" in this manuscript.

L277-279, There was DNA contamination in ROCs, and 40-90% of ROCs were assigned as Firmicutes or Bacteroidetes. Authors should document how to exclude the effect of contaminated DNA in Hare and Rabbit samples.

External reviews were received for this submission. These reviews were used by the Editor when they made their decision, and can be downloaded below.

---

## Round 0.2 · accepted · Accept

· Academic Editor

Accept

Thank you for the thorough response to all of the reviewers' comments.

·

Basic reporting

The authors have dealt with the comments I made in an appropriate manner.

Experimental design

No comment

Validity of the findings

No comment

Additional comments

The authors have dealt with the comments I made in an appropriate manner.

Reviewer 2 ·

Basic reporting

no comment

Experimental design

no comment

Validity of the findings

no comment

Additional comments

The manuscript is well written and shows great information.
I have just found a minor correction:
Line 321 Add “)” after 19 in (14-17, 19

·

Basic reporting

no comment

Experimental design

no comment

Validity of the findings

no comment

Additional comments

L154, My previous comment meant I wonder this part should be "observed ASV", rather than "observed OTU".

External reviews were received for this submission. These reviews were used by the Editor when they made their decision, and can be downloaded below.